# Maternal age-dependent APC/C-mediated decrease in securin causes premature sister chromatid separation in meiosis II

Ibtissem Nabti[1,2,3], Rosanna Grimes[2], Hema Sarna[2], Petros Marangos[2,4,5] & John Carroll[1,2]

Sister chromatid attachment during meiosis II (MII) is maintained by securin-mediated inhibition of separase. In maternal ageing, oocytes show increased inter-sister kinetochore distance and premature sister chromatid separation (PSCS), suggesting aberrant separase activity. Here, we find that MII oocytes from aged mice have less securin than oocytes from young mice and that this reduction is mediated by increased destruction by the anaphase promoting complex/cyclosome (APC/C) during meiosis I (MI) exit. Inhibition of the spindle assembly checkpoint (SAC) kinase, Mps1, during MI exit in young oocytes replicates this phenotype. Further, over-expression of securin or Mps1 protects against the age-related increase in inter-sister kinetochore distance and PSCS. These findings show that maternal ageing compromises the oocyte SAC–APC/C axis leading to a decrease in securin that ultimately causes sister chromatid cohesion loss. Manipulating this axis and/or increasing securin may provide novel therapeutic approaches to alleviating the risk of oocyte aneuploidy in maternal ageing.

[1] Development and Stem Cells Program, Department of Anatomy and Developmental Biology, Monash Biomedicine Discovery Institute, Monash University, Melbourne, VIC 3800, Australia. [2] Division of Biosciences, Department of Cell and Developmental Biology, University College London, London WC1E 6BT, UK. [3] Division of Science, New York University Abu Dhabi, Abu Dhabi, PO Box 129188, UAE. [4] Department of Biological Applications and Technology, University of Ioannina, Ioannina 45110, Greece. [5] Department of Biomedical Research, Institute of Molecular Biology and Biotechnology-Foundation for Research and Technology, Ioannina 45110, Greece. Correspondence and requests for materials should be addressed to I.N. (email: i.nabti@nyu.edu) or to J.C. (email: j.carroll@monash.edu).

The ability of an oocyte to undergo correct chromosome segregation during the two meiotic divisions is essential for production of a healthy viable fetus. It is well known that abnormalities in chromosome segregation in female meiosis increase during maternal aging such that after the age of 35 there is a significant decrease in fertility, an increase in the rate of miscarriage and an increase in the risk of chromosomal anomalies such as Down's syndrome[1–4]. The exponential relationship between maternal age and aneuploidy (chromosome number abnormalities) is illustrated by the finding that by the age of 40 it is estimated that 40–60% of oocytes are aneuploid[2,3,5].

The first meiotic division (MI) is thought to be the origin of most aneuploidy[1]. Oocytes enter MI in fetal life and it is not until a hormonal signal reinitiates meiosis just prior to ovulation that MI is completed. After a brief interkinesis, oocytes progress to metaphase of the second meiotic division (MII) before fertilization triggers the completion of MII and entry into the first embryonic mitosis. The successful completion of the meiotic divisions requires the co-ordinated control of chromosome segregation by tightly regulating the activity of separase, a protease necessary for cleaving the cohesin ring that holds chromosomes together until the correct moment[6,7]. Timely activation of separase is coupled to M-phase exit by the anaphase promoting complex/cyclosome (APC/C)-mediated destruction of the separase inhibitor, securin[8–12].

Tight control of separase in MI is particularly critical because centromeric cohesin needs to be protected to maintain sister chromatid cohesion for MII[11,13,14]. Thus in MI, cohesin cleavage needs to be restricted to the chromosome arms so as to allow for resolution of chiasmata and segregation of homologous chromosomes. This selective cleavage of cohesin on chromosome arms is achieved through a Shugoshin (Sgo2)-mediated protection of centromeric cohesion[9,15–18]. Thus, loss of this protection, or overriding it through unbridled separase activity leads to premature sister chromatid separation in MI due to premature cleavage of centromeric cohesin[9,15–18].

The rapid sequential progression from MI to MII also presents significant challenges for securin-mediated control of separase. Securin is degraded by the APC/C during exit from MI, leaving markedly reduced levels in MII-stage oocytes[19,20]. Maintaining tight control of securin during the MI-to-MII transition is therefore essential to ensure sufficient securin remains in MII so as to inhibit separase and maintain sister chromatid cohesion until fertilization triggers exit from MII[19].

Cohesin is particularly susceptible to aging and chromosome-associated cohesin levels are reduced in oocytes from old mice[21–23]. This loss of cohesin is thought to be the basis of increased aneuploidy caused by maternal aging[3,4] although it is likely to be compounded by other aging-related deficits, including a compromised ability to correct kinetochore-microtubule miss-attachments[24] and a decrease in the ability of the spindle assembly checkpoint (SAC) to detect incorrectly attached chromosomes[25]. The susceptibility to premature loss of cohesin is considered to be due to the fact that cohesin is loaded onto chromosomes as oocytes enter meiosis in fetal life and, as demonstrated by elegant genetic studies in mice, there is no capacity to reload cohesin once it is lost[26]. Thus, an aging-related loss of cohesin due to accumulated insults and cellular damage is thought to contribute to destablization of chiasmata; the cross-over sites responsible for holding homologous chromosomes together in MI[4]. Additional mechanisms that lead to the loss of Sgo2-mediated protection of centromeric cohesin have also been implicated and is supported by the finding that premature sister segregation occur in MI[27] as well as MII[28]. The fact that premature sister separation is seen in MII and is a relatively a common form of aneuploidy in mouse and human oocytes[21,23,28–30] raises the possibility that deficits in cohesin seen in aged oocytes could well be further exacerbated if separase activity was increased in MII oocytes as a result of aberrant securin depletion at exit from MI.

Here, we find that securin levels are reduced specifically in MII oocytes from old mice. The cause appears to be an increased APC/C-mediated securin destruction in old oocytes; an effect that can be phenocopied in young oocytes by inhibiting the SAC component, monopolar spindle 1 (Mps1) Kinase, during the destruction phase. Finally, we show that restoring securin levels or overexpressing Mps1 partially reverses the inter-sister kinetochore distance and decreases the frequency of premature sister chromatid separation (PSCS).

## Results

**Decreased sister cohesion in MII eggs from aged MF1 mice.** Because strain differences have been reported in the susceptibility of oocyte quality to aging[25], we first set out to determine if MII eggs from our laboratory-aged MF1 mice exhibit age-related changes reported in other laboratories. These include chromosome misalignment, increased inter-sister kinetochore distance and PSCS. To analyse chromosome alignment in MII-arrested oocytes from young (1 month old) and old (13–14 month old, hereafter referred to as >1 year old) mice, we generated confocal z-stacks of DNA and microtubules in fixed MII oocytes. The images revealed that 45% of MII oocytes from old mice had evidence of misaligned chromosomes compared to only 15% of young controls (Fig. 1a,b). Next, we used monastrol to induce monopolar spindles which, after staining with Hoechst and CREST, provide an in-situ chromosome spread on which it is possible to measure inter-sister kinetochore distance[31–33]. In MII-arrested eggs from old mice, there was a highly significant 2-fold increase in inter-sister kinetochore distance compared to controls (Fig. 1c,e). Furthermore, the incidence of PSCS in old oocytes was 63% compared to only 8% in young control oocytes (Fig. 1d,e). These data demonstrate that in our MF1 aging model, there is a significant disruption to spindle formation and to sister cohesion, most likely caused by the previously reported age-dependent decrease in cohesin[21,23,24,28,34–36].

**Age-related decrease in securin stability.** The two consecutive meiotic cell divisions create a significant challenge in that cell cycle proteins destroyed on exit from MI are then needed again for controlling entry into and exit from MII. Securin, the inhibitor of separase, is one such protein. Unlike cyclin B1, the level of securin remains low after MI[19,20], such that securin levels in MII oocytes are ~30% of that seen in MI (Supplementary Fig. 1). Despite this relatively low level of securin, we have previously shown that securin is essential for inhibition of separase in MII[19]. Given the increase in inter-sister kinetochore distance and PSCS in old oocytes, we investigated the levels of securin in young and old MII oocytes. Analysis of immunoblots revealed a significant 48% decrease in the amount of securin in old MII oocytes compared to young controls (Fig. 2a,b). To examine whether these differences existed prior to MII, we performed western blots on germinal vesicle (GV) and MI-stage oocytes. In contrast to MII, the endogenous levels of securin in the GV and MI stages were not affected by maternal age (Fig. 2c–f). These data suggest that in old oocytes an increase in securin instability in the MI-to-MII transition leads to a reduced level of securin in MII oocytes.

**Compromised SAC and increased APC/C activity in aged oocytes.** To examine securin destruction during the MI-to-MII transition, we monitored securin-GFP during the period of APC/C activity. As has been reported previously, there was no apparent difference in the timing of the onset of APC/C-mediated

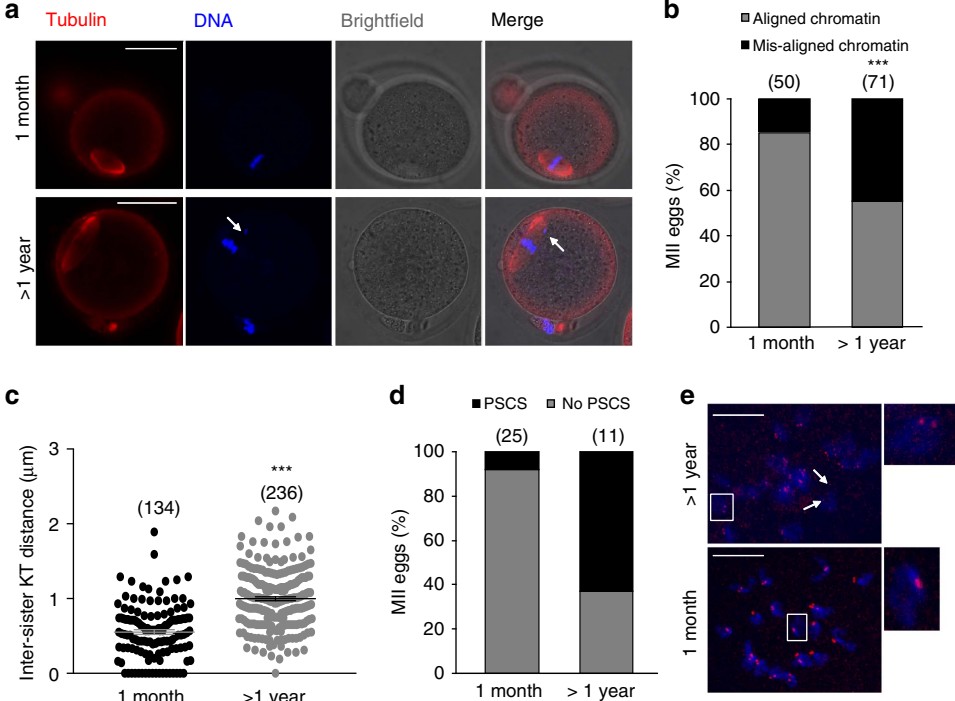

**Figure 1 | Chromosome misalignment and reduced sister chromatid cohesion in MII eggs from aged MF1 mice.** Representative images of immunostaining for DNA in blue and tubulin in red (**a**), and quantification of chromosome misalignment (**b**) in MII eggs from young (1 month) and aged (>1 year) mice. The number of eggs used is shown in parentheses. The arrow points towards misaligned DNA. (**c**) The mean distance between sister kinetochores in MII eggs from 1 month (n = 14 eggs) and >1 year (n = 20 eggs) old mice. The number of kinetochores measured is shown in parentheses. In **b,c**, the results are mean ± s.e.m. ***P < 0.001. P values were calculated with one-sided Student's t-test. (**d**) Rates of PSCS in MII eggs from 1 month- versus >1 year old mice. The number of eggs used is shown in parentheses. (**e**) Representative example of the chromosome spreads assay demonstrating MII eggs from >1 year- (top panel) and 1 month old (bottom panel) mice. DNA is shown in blue and CREST-labelled kinetochores are shown in red. The arrows point towards two-separated sister chromatids, and the insets show the difference in the inter-sister kinetochore distance between eggs from 1 month- and >1 year old mice. Scale bars, 50 μm. Results are from three to four independent experiments involving two to five mice per experimental group.

securin degradation[23] (Supplementary Fig. 2a). However, consistent with the observed decrease in the level of endogenous securin in old oocytes, the rate of securin-GFP destruction was significantly increased in oocytes from old mice compared to young controls (Fig. 3a,b and Supplementary Fig. 2b). A number of factors may explain the mechanism underlying the increased securin degradation in the MI-to-MII transition including an increase in APC/C activity, post-translational modification of securin, or a change in the capacity of the 26S proteasome. Of these possibilities, we have previously found securin phospho-mutants all get degraded at a similar rate (unpublished data) and there is no evidence that the proteasome is limiting in oocytes. In contrast, the APC/C is highly regulated during meiosis and two recent independent studies have shown that APC/C activity at the MI-to-MII transition is restrained by residual SAC activity. In these studies, inhibition of the SAC using Mps1 inhibitors resulted in an increased rate of APC/C substrate degradation during exit from MI[31,37]. In addition, oocytes from old mice have been shown to have a reduced level of SAC components, such as BubR1 (budding uninhibited by benzimidazoles related 1)[25,38]. Therefore, we have asked whether the age-related increase in securin destruction during the MI-to-MII transition and the associated decrease in securin at the MII stage are caused by a decrease in residual SAC activity in old oocytes.

First, given that oocytes from old mice are reported to have decreased levels of BubR1 (refs 25,38) and because compromised SAC activity is usually associated with a loss of SAC components

from the kinetochores, we went on to examine whether the kinetochore level of BubR1 in late MI (11 h post release), just prior to PB1 extrusion, differs in young and old eggs. As an additional positive control, the essential SAC kinase, Mps1, was inhibited by including AZ314631 in the media from 7.5 h after release. Kinetochore BubR1 immunofluorescence was significantly decreased by 39% in old oocytes relative to young controls (Supplementary Fig. 3), while young AZ3146-treated oocytes showed a 73% decrease compared to untreated controls (Supplementary Fig. 3). These finding support the idea that the residual SAC activity during the MI-to-MII transition may be less effective in old oocytes due to a reduced level of BubR1 remaining on the kinetochores, even at this late stage of MI.

To address the potential role of the SAC–APC/C axis in causing the increased rate of securin destruction in old eggs we have asked whether inhibition or activation of the SAC is sufficient to replicate the effects of maternal ageing on young oocytes. If the SAC is the responsible factor, its inhibition at the start of APC/C activity would be predicted to lead to similar rates of securin destruction in young and old oocytes. To test this hypothesis Mps1 was inhibited (as above) so as to prevent SAC activity during exit from MI and securin-GFP destruction was measured. The data confirm the hypothesis and show that inhibition of the SAC increases the rate of securin destruction in young oocytes to levels similar to that of oocytes from old mice (Fig. 3c,d and Supplementary Fig. 2c).

The inhibition of the SAC in both young and old oocytes results in a much sharper onset of destruction compared to

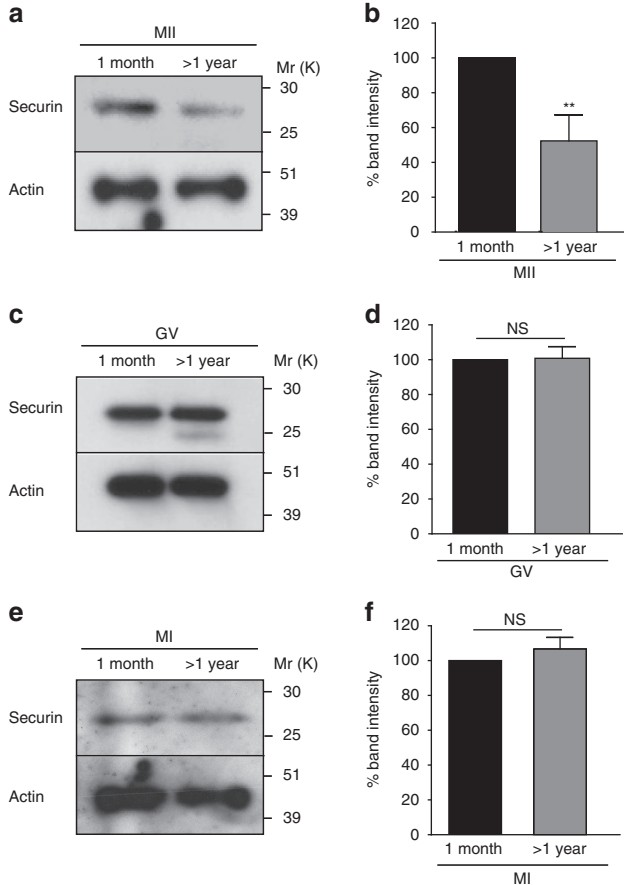

**Figure 2 | Age-related decrease in securin stability during MII but not MI.** Western blots (**a**,**c**,**e**) and densitometric analysis (**b**,**d**,**f**) of oocytes from 1 month- and >1 year old mice for securin during MII arrest (14 h post release from GV, 50 oocytes per lane), GV arrest (20 oocytes per lane) and MI (8 h post release from GV, 20 oocytes per lane). Actin was used as a loading control. Results are mean ± s.e.m. $^{NS}P > 0.05$ and $**P < 0.01$. P values were calculated with one-sided Student's t-test. All western blots were repeated three times using oocytes from two to six mice per experimental group.

untreated oocytes (compare peak of graphs in Fig. 3a,c). To test whether this is simply an effect the Mps1-inhibitor synchronizing the time of onset of destruction, we plotted the securin destruction data normalizing the peak of each trace to start the same point in time (Fig. 3e–g). The data show that despite removing the variability in time of onset of destruction in control oocytes (Fig. 3e), there remains a markedly faster onset of destruction in Mps1-treated oocytes (Fig. 3f). This is clearly illustrated when plotting old control and Mps1 inhibitor-treated oocytes on the same graph; although the maximal rate of destruction is the same, the inflection point is sharper in the inhibitor-treated oocytes leading to a delay in the destruction by about 20 min. These observations reveal that in control conditions in both young and old oocytes the SAC is switched off slowly in a graded manner rather than it being a switch-like mechanism as is imposed in the presence of the Mps1 inhibitor.

Having shown that Mps1 inhibition in young oocytes accelerates securin destruction to rates similar to that seen in old oocytes, we asked whether Mps1 inhibition can also recapitulate the decrease in securin seen in oocytes from old mice (Fig. 2). Western blotting shows that oocytes treated with Mps1-inhibitor during the MI-to-MII transition do indeed have a

40% reduction in securin at the MII stage (Fig. 3h,i). Thus modulating the SAC, and by inference, APC/C activity, during the MI-to-MII transition is sufficient to mimic the effects of maternal ageing on the levels of securin in MII oocytes.

**Rescue of the age-related increase in sister cohesion loss.** Next, we examined if increasing the rate of securin destruction in young oocytes in the MI-to-MII transition can also cause an increase in inter-sister kinetochore distance in young oocytes as is seen in old oocytes[21,23,24,28,36] (Fig. 1). Oocytes were treated with AZ3146-7.5 h after release and allowed to progress to the MII stage at which point they were fixed and processed for in-situ chromosome spreads and CREST-labeling of kinetochores. Measurement of the inter-sister kinetochore distances revealed that inhibition of Mps1 in young oocytes phenocopies the age-related loss in sister-kinetochores attachment by causing a significant (almost 3-fold) increase in the inter-sister kinetochore distance compared to non-treated controls (Fig. 4a,b).

Given these findings, we asked whether expression of exogenous Mps1 could protect against the increased inter-sister kinetochore distance seen in oocytes from old mice. Mps1-GFP cRNA was injected into GV oocytes from young and old mice and the oocytes were matured to the MII stage (14 h). After expression of Mps1-GFP, inter-sister kinetochore distances were significantly reduced in oocytes from old mice and no effect was seen in oocytes from young mice (Fig. 4c,d). Thus, manipulating the SAC–APC/C axis via inhibition or expression of Mps1 not only replicates the changes in securin in old oocytes, it also predictably impacts inter-sister kinetochore distances.

Finally, given that depleting securin in MII oocytes leads to PSCS[19] and that MII-stage oocytes from old mice have reduced levels of securin, we have asked whether increasing the level of securin in oocytes from old mice can protect against the observed decrease in cohesion and increase in PSCS. We first confirmed that injection of securin-GFP cRNA into GV-stage oocytes led to a detectable securin-GFP fluorescence in MII-stage oocytes (Supplementary Fig. 4). The inter-sister kinetochore distance in old oocytes injected with securin-GFP is less than that seen in non-injected old oocytes (Fig. 4d). Increasing the level of securin in young oocytes had no effect on inter-sister kinetochore distances (Fig. 4e) suggesting that securin levels are limiting in old eggs but not young. To check that the ability of securin (and Mps1) expression to reduce inter-sister kinetochore distance was not due to a small number of oocytes showing large inter-sister kinetochore distances, we also analysed the mean inter-sister kinetochore distance/oocyte (Supplementary Fig. 5). This confirmed the analysis performed by measuring individual sister kinetochore pairs across a population of oocytes and supports the conclusion that securin-GFP expression protects against the loss of cohesion in MII-stage oocytes from old mice. Furthermore, the protection of cohesion in securin-GFP-injected oocytes from old mice is accompanied by a 27% decrease in the incidence of PSCS (Fig. 4f).

**Discussion**
The findings presented here demonstrate a new mechanism underlying the effects of maternal age on oocyte quality. We find that MII-stage oocytes from old mice have a reduced level of securin compared to oocytes from young mice. It is proposed that this is caused by an increased rate of securin destruction during the MI-to-MII transition and that this aberrant increase in securin degradation leads to incomplete inhibition of separase resulting in premature loss of sister chromatid cohesion. The finding that restoration of securin levels or increasing Mps1 can partially recover inter-sister kinetochore distance and

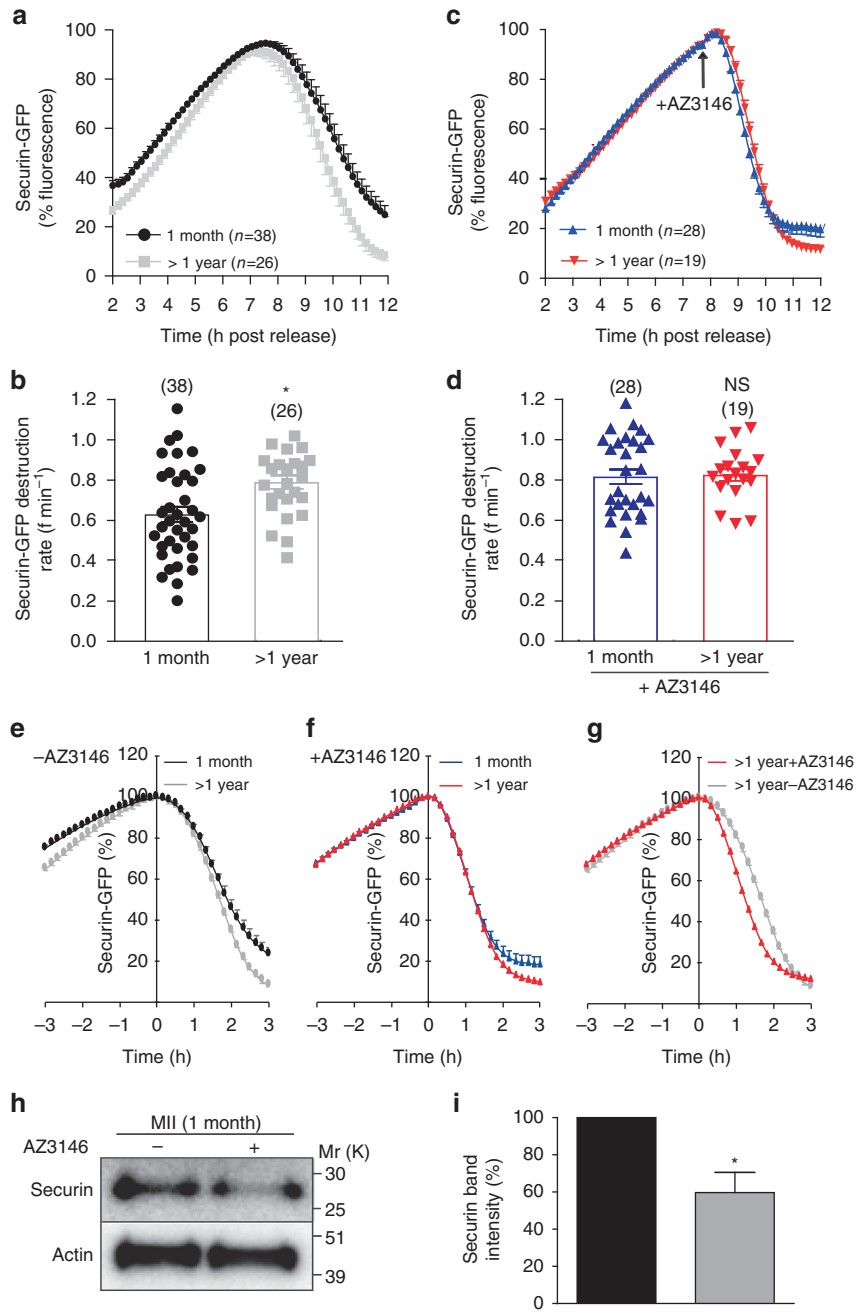

**Figure 3 | Compromised SAC and higher APC/C-mediated securin destruction at the MI exit in oocytes from aged mice.** Time-lapse fluorescence readings (**a,c**) and rates of securin-GFP destruction (**b,d**) in oocytes from 1 month- and >1 year old mice in the absence (**a,b**) or presence (**c,d**) of the Mps1 inhibitor (AZ3146), which was added to the imaging medium at 7.5 h post release from GV arrest (arrow). Number of oocytes used is shown in parentheses. (**e–g**) Re-analysis of the data in **a,c**. The timing of onset of securin-GFP destruction is synchronized by normalizing time 0 to the peak of the curves. Western blot and (**h**) densitometric analysis (**i**) of MII oocytes (100 oocytes per lane, $n = 3$) from 1 month old mice for securin. All oocytes were collected at 14 h post release from GV arrest. The AZ3146-treated oocytes were incubated in the presence of the drug from 7.5 h post release from GV arrest. Actin was used as a loading control. All results are mean ± s.e.m. $^{NS}P > 0.05$ and $*P < 0.05$. P values were calculated with one-sided Student's t-test. Results are representative of two to three independent experiments involving two to five mice per experimental group.

PSCS has implications for future research directions and potential therapeutic approaches to blunting the impact of maternal age on oocyte quality.

Our study identifies the mechanism leading to the observed decrease in securin in oocytes from old mice. The finding that old oocytes destroy APC/C substrates at a faster rate was unexpected but is consistent with recent studies showing that in young

healthy oocytes undergoing MI exit, the APC/C is restrained by SAC components[31,37]. Here, we show that this SAC-mediated restraint appears to be compromised in oocytes from old mice. The evidence supporting a role for the SAC in the increased APC/C-mediated securin destruction in old oocytes includes (i) the fact that inhibiting the SAC during the MI-to-MII transition in young control oocytes leads to an increased rate of

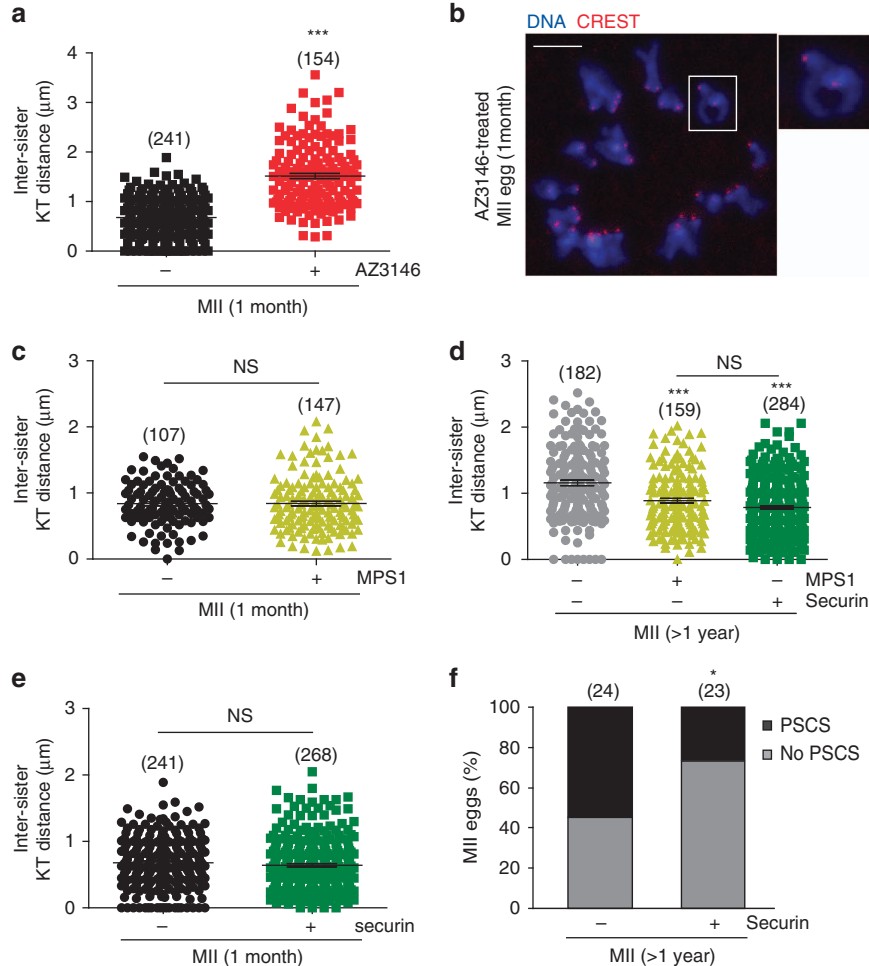

**Figure 4 | Rescue of the age-related increase in inter-sister kinetochore distances and PSCS frequency by over-expression of Mps1 or securin.**
(**a**) Mean inter-sister kinetochore distance in MII eggs from 1 month old mice in the absence (*n* = 21 eggs) or presence (*n* = 20 eggs) of AZ3146. The AZ3146-treated oocytes were incubated with the drug from 7.5 h post release from GV arrest. (**b**) Representative example of the chromosome spreads demonstrating the increase in inter-sister kinetochore distances (inset) in the AZ3146-treated MII oocytes. DNA is shown in blue and CREST-labelled kinetochores are shown in red. Scale bar, 10 μm. (**c**) Mean inter-sister kinetochore distances in MII eggs from 1 month old mice in the absence (*n* = 12 eggs) or presence (*n* = 23 eggs) of Mps1-GFP. (**d**) Mean inter-sister kinetochore distances in MII eggs from >1 year old mice in the absence (*n* = 20 eggs) or presence of either Mps1-GFP (*n* = 20 eggs) or securin-GFP (*n* = 27 eggs). (**e**) Mean inter-sister kinetochore distances in MII eggs from 1 month old mice in the absence (*n* = 21 eggs) or presence (*n* = 29 eggs) of securin-GFP. All MII eggs were fixed at 14 h post release from GV arrest. Mps1-GFP and securin-GFP cRNA were microinjected at the GV stage. The number of sister-kinetochores measured is shown in parentheses. (**f**) Rates of PSCS in MII eggs from >1 year old mice in the presence or absence of securin. The number of eggs used is shown in parentheses. All results are mean ± s.e.m. NS*P* > 0.05, *\**P* < 0.05 and \*\*\**P* < 0.001. *P* values were calculated with one-sided Student's *t*-test. Results are representative of two to four independent experiments involving two to five mice per experimental group.

securin destruction and reduced MII levels of securin, similar to that seen in old oocytes, and (ii) the observation that the level of kinetochore-localized BubR1 during MI exit is decreased compared to young controls. Thus, our data strongly implicate aging-related decreases in SAC function to be the cause of increased APC/C-mediated securin destruction in the MI-to-MII transition.

The role of the SAC in aging-related oocyte defects has been subject to much experimentation. Initial studies indicated that the SAC was unaffected in old oocytes because the time of polar body formation is the same in young and old oocytes; an ineffective SAC should allow for faster progression into MI exit[39,40]. However, more recent studies have revealed that the SAC is less effective in response to nocodazole[24] or DNA damage-induced[38] chromosome anomalies. This subtle, context dependent decrease in SAC function in old oocytes, together

with the recent discovery that the SAC restrains the active APC/C during the MI-to-MII transition[37], is consistent with our observation that old oocytes have an increased APC/C-mediated destruction of securin.

One consequence of increased APC/C-targeted destruction of securin in the MI-to-MII transition is that old oocytes have a 48% decrease in the levels of securin compared to young controls and an accompanying increase in inter-sister kinetochore distance and increase in PSCS. In the MI-to-MII transition, the control of securin levels appears to be of critical importance because, unlike cyclin B1, which increases after MI, securin remains 3-fold lower in MII oocytes compared to immature GV and MI-stage oocytes, where it also acts as a buffer of APC/C activity to maintain levels of cyclin B1 (refs 19,20). Despite this relatively low level of securin in MII oocytes, we have previously shown that securin is both necessary

and sufficient for separase inhibition at MII[14,19]. Thus, given the relatively low levels of securin at the MII stage, the further reduction of securin in old MII oocytes may breach the threshold necessary to maintain a tight grip on separase activity, resulting in compromised cohesion and increased aneuploidy[24,28]. Interestingly, securin knock-out mice show sub-fertility with a litter size approximately half that of wild-type controls[41]. This is consistent with our study and it would be interesting to investigate the origin of this fertility loss to determine whether it is caused by an increase in aneuploidy as well as to investigate whether there is any compensation acting to inhibit separase via other pathways[42].

Our data provide evidence supporting a role for the SAC–APC/C axis and decreased securin in the loss of cohesion in MII oocytes. Firstly, inhibition of the SAC in young oocytes undergoing the MI-to-MII transition leads to loss of cohesion, similar to that seen in old oocytes. Secondly, expressing Mps1 in old oocytes partially reverses the increase in inter-sister kinetochore distance and finally, that expression of exogenous securin to enhance separase inhibition in MII oocytes leads to a partial rescue of inter-sister kinetochore distances and PSCS in old oocytes. These data are consistent with an important role for SAC-mediated restraint of APC/C-mediated destruction of securin so as to maintain sufficient levels to provide a complete inhibition of separase during the transition into MII. The partial nature of the ability to rescue the phenotype with exogenous Mps1 or securin suggest that it is likely that other factors emanating from MI events also contribute to the decrease in cohesion seen in old MII oocytes.

Previous studies in mouse and human oocytes have demonstrated a maternal aging-associated loss of cohesion in both MI and MII, but a particular increase in PSCS in MII oocytes has also been reported[5,28]. The aging-related loss of cohesion in MII is attributed to a decrease in centromeric cohesin caused by a loss of Sgo2-mediated cohesin protection[9,23,28] or due to a more generic loss of cohesin function due to age and associated environmental insults[3,43]. These factors will undoubtedly contribute to the defects in cohesion seen in old MII oocytes but our study shows that these underlying deficits may be amplified if securin levels are reduced and thereby allow for incomplete inhibition of separase. As such, the integrity of cohesin in maternal aging of MII oocytes is likely compromised via a number of modalities.

Our findings suggest novel approaches to therapeutically improving sister chromatid cohesion in MII and therefore decreasing the rates of aneuploidy seen in oocytes from old mothers. Two approaches have been shown to decrease inter-sister kinetochore distances in old oocytes: first, increasing the oocyte levels of Mps1 to strengthen the SAC during MI exit and secondly, restoring the levels of securin to enhance separase inhibition in MII oocytes. These approaches both improve cohesion but they target the same pathway so are unlikely to be additive. However, combining one of these approaches with increasing Sgo2 to increase protection of centromeric cohesin may provide a highly effective combinatorial approach to improving the fidelity of chromosome number of oocytes in cases of advanced maternal age.

## Methods

**Animals.** All oocytes were collected from 1 month or >1 year (13–14 months) old female MF1 mice (Harlan). All experiments were preformed according to licensed procedures under a Home Office Project Licence to JC.

**Oocytes collection and culture.** For GV oocytes, females were super-ovulated by intraperitoneal injection of 7.5 IU of pregnant mares' serum gonadotrophin (PMSG, Intervet). Mice were killed by cervical dislocation at 46–48 h after PMSG injection. The ovaries were removed and immediately transferred to dissection medium, consisting of M2 medium supplemented with 200 μM IBMX, to keep the oocytes arrested at the GV stage. The cumulus-enclosed oocytes were isolated by mechanical perforation of the ovaries with a 27-gauge needle. The cumulus cells were removed by repeated mouth pipetting, using narrow-bore glass Pasteur pipettes. To obtain MII-stage oocytes, mice were first injected with 7.5 IU PMSG, followed by 5 IU of human chorionic gonadotrophin (hCG) 48 h later. 14 h after hCG injection, mice were killed as before and oviducts were dissected and transferred into M2 medium at 37 °C. Masses of cumulus-enclosed MII oocytes were released from oviducts using forceps. Overall, 300 IU ml⁻¹ hyaluronidase was then added to the medium to remove the cumulus cells; oocytes were collected by mouth pipetting and washed in M2 medium. For longer-term incubation, GV and MII oocytes were cultured in M16 in a 5% $CO_2$ humidified incubator at 37 °C.

**Treatment with inhibitors.** For Mps1 inhibition, oocytes were incubated in culture media (M2 or M16) containing 2 μM AZ3146 (Santa Cruz Biotechnology).

**Microinjection and imaging.** All microinjections of GV-stage oocytes were performed in M2 medium on the stage of an inverted microscope (Leica DM IRB, Leica, Wetzlar, Germany). Briefly, fabricated micropipettes were inserted into oocytes using the negative capacitance overcompensation facility on an electrophysiological amplifier (World Precision Instruments, UK) while immobilized using a holding pipette (Hunter Scientific). A precise injection volume (2–5% of the total egg volume) was achieved using a Pneumatic PicoPump. Epi-fluorescence images of oocytes incubated in M2 medium at 37 °C were recorded using a (20 × 0.75 NA) objective and a Princeton Instruments MicroMax interline cooled CCD camera (Roper Scientific, Buckinghamshire, UK). GFP-tagged securin was imaged using a fluorescein isothiocyanate (FITC) filter set at band pass 450–490 nm for excitation, dichroic mirror 510 nm and band pass 520 nm for emission. Metamorph (MM) and metafluor (MF) software 6.1 (Universal Imaging, PA, USA) were used for image capture and data analysis. Oocytes were imaged every 10 min to minimize photobleaching and photodamage.

**Immunofluorescence.** Oocytes were fixed and permeabilized in PHEM buffer (60 mM Pipes, 25 mM Hepes, 10 mM EGTA and 2 mM $MgCl_2$) containing 4% paraformaldehyde and 0.5% Triton X-100 and then labelled with mouse anti–β-tubulin (T4026; 1:1,000; Sigma-Aldrich), and 10 μg ml⁻¹ Hoechst 33342 (Sigma-Aldrich). Serial z sections of fixed oocytes in PBS were acquired at room temperature using a Plan Apochromat 63 × , 1.4 NA oil differential interference contrast objective and a laser-scanning confocal microscope imaging system (LSM 510 META; Carl Zeiss) with the following band pass emission filters in nm 385–470 (Hoechst 33342) and 585–615 (Alexa Fluor 546). Z sections were analysed and projected onto a single plane using the LSM image browser (Carl Zeiss).

**Chromosome spreads.** Chromosome spreads were performed as previously described[31–33]. In brief, spindles were collapsed using a 90-min pulse of 200 μM monastrol (EMD Millipore). Oocytes were fixed and permeabilized in PHEM buffer, labelled with calcinosis, Raynaud's phenomenon, oesophageal dysmotility, sclerodactyly, and telangiectasia (CREST) serum, a human centromere antiserum, (1:300; kind gift from Bill Earnshaw, University of Edinburgh, UK), rabbit anti-BubR1 (1:300; kind gift from Stephen Taylor, University of Manchester, UK), 10 μg ml⁻¹ Hoechst 33342 and mounted on slides in PBS. Serial z sections were acquired at 1-μm intervals at room temperature using a Plan Apochromat 63 × , 1.4 NA oil differential interference contrast objective and a laser-scanning confocal microscope imaging system (LSM 510 META) with the following band pass emission filters (nm): 385–470 (Hoechst 33342), 505–530 (Alexa Fluor 488) and 585–615 (Alexa Fluor 546). Z sections were analysed and projected onto a single plane using the LSM image browser.

**Complementary RNA.** The cRNA for GFP-tagged securin and Mps1 were prepared from the T3-promoter of a pRN3-GFP vector, using T3 mMESSAGE mMACHINE kit (Ambion). The cRNA was then polyadenylated and purified in nuclease-free water to a concentration of ∼1 μg μl⁻¹ before microinjection.

**Western blotting.** Oocytes were washed in phosphate-buffered saline (PBS) with 1% polyvinylpyrrolidone (PVP) solution and then heated at 95 °C for 5 min with 5 × sample buffer. Proteins were fractionated at 200 mV for 50 min on an X Cell II blot Module (Invitrogen, UK) using a 4–12% NuPage Bis-Tris pre-cast gel (Invitrogen) and MOPS running buffer. Proteins were blotted onto polyvinylidened fluoride membranes (PVDF) for 1 h 30 min at 100 mV. Anti-securin (ab3305, 1:1,000, AbCam, UK) and anti-beta actin (ab3280, 1:400, AbCam, UK) were used for western blotting. For primary antibody detection we used HRP-conjugated anti-mouse/rabbit secondary antibodies (Sigma, UK). Standard enhanced chemiluminescence (ECL) techniques (Amersham Biosciences, UK) were used for secondary antibody detection according to manufacturer's instructions. Actin was always used as a loading control and the densitometric analysis of the blots involved the measurement of the intensity of each band, which was then normalized against the relevant actin loading control. The whole gel images of western blots are presented in Supplementary Fig. 6.

**Data analysis.** The rate of securin-GFP destruction per hour was calculated according to the formula $(Fl_1–Fl_2)/(t_2 − t_1)$. $t_1$ and $t_2$ are the time points that

correspond to Fl$_1$ and Fl$_2$, respectively. Background fluorescence was subtracted from cell fluorescence.

To measure the inter-sister kinetochore distance, we used ImageJ software (National Institutes of Health). In brief, a linescan was drawn across each pair of CREST-labelled sister kinetochores. The inter-sister kinetochore distance was then measured as the distance between the two peaks of fluorescence, which represent the centre of each kinetochore. However, when the sister kinetochores were not in the same plane of focus, we used the Pythagorean theorem to calculate the actual inter-sister kinetochore distance.

For BubR1 levels measurement, we used ImageJ to measure the fluorescence of each kinetochore from a single $z$ section. The CREST image was used for localization comparison. As background, we used the mean fluorescence of an area surrounding the DNA, which was then subtracted from the kinetochore fluorescence.

**Statistics.** All experiments were repeated at least twice with individual sample sizes of 10–40 oocytes, which is sufficient to determine the magnitude of the effect. For each experiment, oocytes were collected from at least two randomly chosen animals and in all cases oocytes from individual mice were pooled and randomly assigned to groups. Where appropriate (such as measurement of inter-kinetochore distances), samples were analysed blind with experimental groups being de-identified and decoded after analysis. For all data, the appropriate statistical test was used and performed using Prism software (GraphPad Software). Individual variance in the datasets is provided by showing each experimental point and standard error bars.

**Data availability.** The data supporting the findings of this study are available within the article and its Supplementary Information file, or from the corresponding authors on a reasonable request.

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

## Acknowledgements

This work was supported by a Medical Research Council Program Grant and an Australian Research Council Discovery Project Grant to J.C.

## Author contributions

I.N. and J.C. designed the study. I.N. performed the majority of the experiments with contributions from R.G., H.S. and P.M. I.N. analysed the data. I.N. and J.C. wrote the manuscript.

## Additional information

**Competing interests:** The authors declare no competing financial interests.

