## [Peer Review File · Nature Communications]

Editorial Note: Parts of this peer review file have been redacted as we could not obtain permission to publish the reports of reviewer #2 .

Reviewers' Comments:

Reviewer #1 (Remarks to the Author)

The present manuscript by Nabti and colleagues aims at clarifying why oocytes from older mice missegregate sister chromatids at higher rates than younger mice. The authors found a decrease in Securin levels in oocytes from older mice, which is likely due to increased APC activity and thereby a failure to properly maintain Separase inactive during the meiosis II arrest. The authors propose that this leads to increased interkinetochore distances and precocious loss of centromeric cohesion leading to the observed missegregations.

Overall, the study is well executed and can be published once the following points have been addressed:

The phrase that "APC/C activity can be normalized by Mps1 inhibition (abstract and page 5)" is misleading, because not old oocytes are somehow rescued to behave like young oocytes, but young oocytes show the same defects as old oocytes, upon Mps1 inhibition. I think this should be formulated differently. (for example: "SAC inhibition in young oocytes results in increased APC/C activity, such as observed in old oocytes")

Loss of Securin protein in MII oocytes: Securin levels in MII oocytes are very low, therefore western blots may be close to detection level, and small variations in protein levels may look much more important. The western blot in MII oocytes (Figure 2) was only done twice and should be repeated a third time to be sure that this is not due to fluctuations in protein detection. Alternatively, are background bands visible on the western blot at around the same size as Securin, which show that there is indeed no difference in blotting efficiency at that molecular size?

As far as I know the inhibitor AZ3146 has not been used in mouse oocytes to inhibit Mps1. The authors should show that the inhibitor indeed overrides the checkpoint in meiosis I, to be sure that the drug is functional to inhibit Mps1 in oocytes. Alternatively the authors should cite references where this drug has been used in oocytes.

In the discussion the authors state that Securin does not re-accumulate after meiosis I and that low levels of Securin remaining in meiosis I oocytes are important for Separase inhibition in meiosis II. As far as I know it has not been shown that remaining Securin levels in meiosis II are indeed "left over" from meiosis I and not newly translated as oocytes enter meiosis II. Even though levels are low they may have been newly synthesized.

Minor point: Figure 2, numbering is not correct in the text (page 7)

Reviewer #3 (Remarks to the Author)

This manuscript investigates one mechanistic aspect of premature sister chromatid separation in meiosis II of mouse oocytes. The major findings support the conclusion that old oocytes have lower levels of securin protein in MII, which is at least in part causative of premature sister chromatid separation. Further data indicate that the kinetics of securin degradation during anaphase I dictate the level of securin in MII and that the Mps1 kinase partly controls the kinetics.

The conclusions are interesting and this is an important topic. Mostly, the data support the conclusions but there are some issues that should be addressed before it would be appropriate to support publication.

Main concerns:

(1) Abstract. "This increased APC/C activity... can be normalized by...inhibiting Mps1." This is strangely worded. It is not increased securin degradation in the old eggs that is corrected by MPS1 inhibition - degradation actually gets faster in the old eggs with the inhibitor.

(2) General point. Through the manuscript they claim the data show APC/C activity is higher in the old eggs during anaphase of MI. This is not shown, just assumed based on more securin degradation. Clearly the data are consistent with the likely possibility that APC/C activity is different in the old eggs, but this is not directly shown. It remains possible that another aspect of securin degradation is affected, for example a securin post-translational modification, securin localization, proteasome regulation. Since other possible factors have not been ruled out, and there is no direct assessment of APC/C activity, the wording needs to be changed throughout to align with the data - i.e. they show that more securin is degraded; mechanism not resolved at this time.

(3) Fig 1b. Significance should be calculated.

(4) Fig 2. The blots are not entirely convincing and the exposures appear to be saturated in some cases. It is important to perform technical replicates of the blots and ensure that the signals are in the linear range, or else use a quantitative method that is completely linear. Three experimental repeats are needed and standard deviation should be shown.

(5) Fig 3. It is not clear if the time-lapse microscopy of securin-GFP levels used z-stacks or single plane of focus. It is important to confirm that the depth of field was sufficient to capture the entire securin signal in each egg/time-point.

(6) Fig 3a-d. Comparing the old mice in 3a,c with or without the MPS1 inhibitor, there appear to be important differences in securin degradation kinetics: (i) increased lag (delayed initiation of degradation) with inhibitor; (ii) faster rate of degradation with inhibitor (the slopes of the curves are quite different in a and c). It is not clear how this translates to equal degradation rates reported in panels b,d. How can this be explained? It seems to me that Mps1 is still restraining the degradation of securin in the old oocytes; it just seems to be having a greater impact in the young oocytes.

(7) Fig 3e,f. There ought to be 3 experimental repeats and technical repeats of the blots that are exposed in a linear range (or quantified using a different method), as per the comments about Fig 2.

(8) Fig 3g,h. BubR1 fluorescence should be reported relative to an internal control. Probably the CREST signal would be the best option with the existing data. Are these z-stacks? Is the entire signal collected?

(9) Fig 4f. Significance should be calculated (n seems low).

RESPONSE TO REFEREES

Manuscript number: NCOMMS-16-11150

Revision submission date: 28/11/2016

Referees requests are in standard font. Responses in italics. Modifications to the manuscript have been thorough and have improved the quality of the manuscript.

Reviewer #1

The present manuscript by Nabti and colleagues aims at clarifying why oocytes from older mice missegregate sister chromatids at higher rates than younger mice. The authors found a decrease in Securin levels in oocytes from older mice, which is likely due to increased APC activity and thereby a failure to properly maintain Separase inactive during the meiosis II arrest. The authors propose that this leads to increased interkinetochore distances and precocious loss of centromeric cohesion leading to the observed missegregations.

Overall, the study is well executed and can be published once the following points have been addressed:

The phrase that "APC/C activity can be normalized by Mps1 inhibition (abstract and page 5)" is misleading, because not old oocytes are somehow rescued to behave like young oocytes, but young oocytes show the same defects as old oocytes, upon Mps1 inhibition. I think this should be formulated differently. (for example: "SAC inhibition in young oocytes results in increased APC/C activity, such as observed in old oocytes")

Response: *Agreed, this is unclear. We have re-formulated the sentence as requested.*

Loss of Securin protein in MII oocytes: Securin levels in MII oocytes are very low, therefore western blots may be close to detection level, and small variations in protein levels may look much more important. The western blot in MII oocytes (Figure 2) was only done twice and should be repeated a third time to be sure that this is not due to fluctuations in protein detection. Alternatively, are

background bands visible on the western blot at around the same size as Securin, which show that there is indeed no difference in blotting efficiency at that molecular size?

Response: *There is a misunderstanding here. The Western blots in MII oocytes were repeated 3 times. The only blot that was done twice was the one using MI oocytes, which shows no change with age. We have now performed an additional replicate and the new data is included in the revised Figure 2. Unfortunately there are no background bands that can be used as controls for blotting efficiency. We achieve this through reblotting for actin and normalizing the securin bands against the actin controls.*

As far as I know the inhibitor AZ3146 has not been used in mouse oocytes to inhibit Mps1. The authors should show that the inhibitor indeed overrides the checkpoint in meiosis I, to be sure that the drug is functional to inhibit Mps1 in oocytes. Alternatively the authors should cite references where this drug has been used in oocytes.

Response: *We have referenced the ability of AZ3146 to override the checkpoint in oocytes as shown by us previously (Nabti et al., Dual-mode regulation of the APC/C by CDK1 and MAPK controls meiosis I progression and fidelity. J Cell Biol 204, 891-900, 2014 (see Figures 3 and 4)).*

In the discussion the authors state that Securin does not re-accumulate after meiosis I and that low levels of Securin remaining in meiosis I oocytes are important for Separase inhibition in meiosis II. As far as I know it has not been shown that remaining Securin levels in meiosis II are indeed "left over" from meiosis I and not newly translated as oocytes enter meiosis II. Even though levels are low they may have been newly synthesized.

Response: *We have rephrased this sentence. It is correct that we cannot state whether the securin in MII oocytes is 'left over' from MI or is newly synthesised in the MI-MII transition, or both. Our intention in this sentence is not to define whether the securin is newly synthesised, rather we are trying to make the simple point that securin is responsible for separase inhibition at MII as shown previously (Nabti et al., Securin and not CDK1/cyclin B1 regulates sister chromatid disjunction during meiosis II in mouse eggs. Dev. Biol. 321, 379-386 (2008)).*

Minor point: Figure 2, numbering is not correct in the text (page 7)

Response: *Thank you, this has now been corrected.*

Reviewer #3

This manuscript investigates one mechanistic aspect of premature sister chromatid separation in meiosis II of mouse oocytes. The major findings support the conclusion that old oocytes have lower levels of securin protein in MII, which is at least in part causative of premature sister chromatid separation. Further data indicate that the kinetics of securin degradation during anaphase I dictate the level of securin in MII and that the Mps1 kinase partly controls the kinetics. The conclusions are interesting and this is an important topic. Mostly, the data support the conclusions but there are some issues that should be addressed before it would be appropriate to support publication.

Main concerns:

Abstract. "This increased APC/C activity... can be normalized by...inhibiting Mps1." This is strangely worded. It is not increased securin degradation in the old eggs that is corrected by MPS1 inhibition - degradation actually gets faster in the old eggs with the inhibitor.

Response: *Yes, agreed, we have re-formulated the sentence as requested. We were trying to say that in the presence of MPS1 Inhibitor securin destruction is similar in old and young eggs.*

General point. Through the manuscript they claim the data show APC/C activity is higher in the old eggs during anaphase of MI. This is not shown, just assumed based on more securin degradation. Clearly the data are consistent with the likely possibility that APC/C activity is different in the old eggs, but this is not directly shown. It remains possible that another aspect of securin degradation is affected, for example a securin post-translational modification, securin localization, proteasome regulation. Since other possible factors have not been ruled out, and there is no direct assessment of APC/C activity, the wording needs to be changed throughout to align with the data - i.e. they show that more securin is degraded; mechanism not resolved at this time.

Response: *Fair point - we have changed the wording as requested. In the discussion we consider any evidence supporting other possibilities as well as the supporting evidence for a role of the APC/C. As noted by the referee the balance of evidence supports the APC/C mediating the effect, particularly because manipulating the SAC generates the predicted response. However, definitive proof of an effect specifically on the APC/C would require a direct measure of the rate of ubiquitination and this is just not possible in the small numbers of mouse oocytes available. I trust this change in emphasis and extra explanation is in line with the reviewer's expectations.*

Fig 1b. Significance should be calculated.

Response: *Significance has been calculated and added to the figure*

Fig 2. The blots are not entirely convincing and the exposures appear to be saturated in some cases. It is important to perform technical replicates of the blots and ensure that the signals are in the linear range, or else use a quantitative method that is completely linear. Three experimental repeats are needed and standard deviation should be shown.

Response: *All blots are repeated at least three times now and SEMs are shown on the graphs. I agree some actin blots look overexposed and made it into the Figs. We have revisited some of the blots and used shorter exposure times for the figures and analysis. Also, for Western analysis, we have previously refined our methods for measuring securin, including ensuring our approach uses analysis in the linear range of the detection method (Marangos and Carroll Nat Cell Biol). For additional reassurance, if we were normalising against saturated actin blots, it would have the effect of decreasing the magnitude of the effect on the levels of securin.*

Fig 3. It is not clear if the time-lapse microscopy of securin-GFP levels used z-stacks or single plane of focus. It is important to confirm that the depth of field was sufficient to capture the entire securin signal in each egg/time-point.

Response: *Securin-GFP imaging was performed using conventional microscopy and not a confocal. We therefore collect fluorescence from the entire oocyte.*

Fig 3a-d. Comparing the old mice in 3a,c with or without the MPS1 inhibitor, there appear to be important differences in securin degradation kinetics: (i) increased lag (delayed initiation of degradation) with inhibitor; (ii) faster rate of degradation with inhibitor (the slopes of the curves are quite different in a and c). It is not clear how this translates to equal degradation rates reported in panels b,d. How can this be explained? It seems to me that Mps1 is still restraining the degradation of securin in the old oocytes; it just seems to be having a greater impact in the young oocytes.

Response: In the control conditions without the MPS1 inhibitor the time of onset of securin destruction in individual oocytes varies over a window of around 1 hour so when it is averaged, the shape of the curve at the top of the graph is very different to that in the presence of the SAC inhibitor. The effect of MPS1 inhibitor is to inhibit the SAC simultaneously in all oocytes leading to a simultaneous switch in all oocytes from SAC active, to SAC inactive. Hence the top of the MPS1 inhibitor curve has a much sharper inflection.

As an alternative analysis of the data we have synchronized the timing of onset of the untreated oocytes by normalising time 0 to the peak of the curve and asked whether this changes the kinetics of the inflection point. Interestingly, this analysis reveals that there is still a significant difference in the kinetics at the top of the curve. In the untreated oocytes there remains a significant delay in the inflection point and the time it takes to reach the maximal rate of destruction (Fig 3e) compared to the MPS1 inhibitor-treated oocytes (Fig 3f). This suggests that in control conditions the mechanism of switching off the SAC is a gradual process rather than a switch-like mechanism. Finally, plotting the old oocytes on the same graph as MPS1 inhibitor-treated old oocytes (Fig 3g) shows the time-difference in the onset of destruction (approximately 20 minutes) and that the maximal rate of destruction, as shown by the slope of the line, is similar.

Given that this form of analysis accounts for variability between oocytes as well as allowing an improved approach to comparing treated and non-treated oocytes, we have opted to present these graphs in the revised Figure 3 of the manuscript and have adapted the text to explain the results.

Fig 3e,f. There ought to be 3 experimental repeats and technical repeats of the blots that are exposed in a linear range (or quantified using a different method), as per the comments about Fig 2.

Response: Apologies if this was not clear. As for comments regarding Fig 2, the westerns are repeated 3 times, and the results are from 3 independent experiments.

Fig 3g,h. BubR1 fluorescence should be reported relative to an internal control. Probably the CREST signal would be the best option with the existing data. Are these z-stacks? Is the entire signal collected?

Response: This would be ideal but it as has been reported previously, the CREST signal in old eggs is not as robust as in young eggs. As such we have performed the experiments contemporaneously in identical conditions for labeling and imaging in order to make the comparisons. To identify an internal control we would need to be sure that there was no change in young and old eggs.

Yes, these are Z-stacks and the entire signal is collected. We have moved this data to the supplementary figures because it is complementary to the key findings.

Fig 4f. Significance should be calculated (n seems low).

Response: *we have repeated the experiment one more time in order to increase the numbers. The significance has been calculated and added to the figure.*

Reviewers' Comments:

Reviewer #1 (Remarks to the Author)

From my side the authors have addressed my concerns in a satisfying manner to allow publication.

Reviewer #3 (Remarks to the Author)

This revised manuscript adequately addresses the concerns raised previously. The data now provide good evidence for the conclusions drawn and in sum the work is an important contribution to the field. I have no remaining reservations.